# Sex-specific innate immune selection of HIV-1 in utero is associated with increased female susceptibility to infection

Emily Adland[1,25], Jane Millar[1,2,25], Nomonde Bengu[3], Maximilian Muenchhoff[4,5], Rowena Fillis[3], Kenneth Sprenger [3], Vuyokasi Ntlantsana[6], Julia Roider[5,7], Vinicius Vieira[1], Katya Govender[8], John Adamson[8], Nelisiwe Nxele[2], Christina Ochsenbauer [9], John Kappes[9,10], Luisa Mori[1], Jeroen van Lobenstein[11], Yeney Graza[12], Kogielambal Chinniah[13], Constant Kapongo[14], Roopesh Bhoola[15], Malini Krishna[15], Philippa C. Matthews [16], Ruth Penya Poderos[17], Marta Colomer Lluch [17], Maria C. Puertas [17], Julia G. Prado[17], Neil McKerrow[12], Moherndran Archary[18], Thumbi Ndung'u[2,8,19], Andreas Groll[20], Pieter Jooste [21], Javier Martinez-Picado [17,22,23], Marcus Altfeld [24] & Philip Goulder[1,2,8,19✉]

Female children and adults typically generate more efficacious immune responses to vaccines and infections than age-matched males, but also suffer greater immunopathology and auto-immune disease. We here describe, in a cohort of > 170 in utero HIV-infected infants from KwaZulu-Natal, South Africa, fetal immune sex differences resulting in a 1.5–2-fold increased female susceptibility to intrauterine HIV infection. Viruses transmitted to females have lower replicative capacity ($p = 0.0005$) and are more type I interferon-resistant ($p = 0.007$) than those transmitted to males. Cord blood cells from females of HIV-uninfected sex-discordant twins are more activated ($p = 0.01$) and more susceptible to HIV infection in vitro ($p = 0.03$). Sex differences in outcome include superior maintenance of aviraemia among males ($p = 0.007$) that is not explained by differential antiretroviral therapy adherence. These data demonstrate sex-specific innate immune selection of HIV associated with increased female susceptibility to in utero infection and enhanced functional cure potential among infected males.

[1] Department of Paediatrics, University of Oxford, Oxford, UK. [2] HIV Pathogenesis Programme, The Doris Duke Medical Research Institute, University of KwaZulu-Natal, Durban, South Africa. [3] Umkhuseli Innovation and Research Management, Pietermaritzburg, South Africa. [4] Max von Pettenkofer Institute, Virology, National Reference Center for Retroviruses, Faculty of Medicine, LMU München, Munich, Germany. [5] German Center for Infection Research (DZIF), Partner site Munich, Munich, Germany. [6] Department of Medicine, University of KwaZulu-Natal, Durban, South Africa. [7] Department of Infectious Diseases, Ludwig-Maximilians-University, Munich, Germany. [8] Africa Health Research Institute, Durban, South Africa. [9] Department of Medicine, University of Alabama at Birmingham, Birmingham, AL, USA. [10] Birmingham Veterans Affairs Medical Center, Research Service, Birmingham, AL 35233, USA. [11] Stanger Hospital, KwaDukuza, KwaZulu-Natal, South Africa. [12] KwaZulu-Natal Department of Health, Pietermaritzburg, South Africa. [13] Mahatma Gandhi Memorial Hospital, Phoenix, KwaZulu-Natal, South Africa. [14] Queen Nandi Regional Hospital, Empangeni, KwaZulu-Natal, South Africa. [15] Edendale Hospital, Pietermaritzburg, KwaZulu-Natal, South Africa. [16] Nuffield Department of Medicine, University of Oxford, Oxford, UK. [17] IrsiCaixa AIDS Research Institute, Germans Trias i Pujol Research Institute (IGTP), Badalona, Spain. [18] Department of Paediatrics, University of KwaZulu-Natal, Durban, South Africa. [19] Ragon Institute of MGH, MIT, and Harvard, Cambridge, Massachusetts, USA. [20] TU Dortmund University, Faculty of Statistics, Vogelpothsweg 87, 44227 Dortmund, Germany. [21] Department of Paediatrics, Kimberley Hospital, Northern Cape, South Africa. [22] University of Vic-Central University of Catalonia (UVic-UCC), Catalonia, Spain. [23] Institució Catalana de Recerca i Estudis Avançats (ICREA), Barcelona, Spain. [24] Virus Immunology Unit, Heinrich-Pette-Institut, Hamburg, Germany. [25] These authors contributed equally: Emily Adland, Jane Millar. ✉email: philip.goulder@paediatrics.ox.ac.uk

Darwinian sexual selection describes the divergent evolutionary forces driving within-specifies sex differences in appearance and behaviour designed to optimise reproductive success, sometimes compromising survival[1]. An important feature of sexual dimorphism is the immune response, with greater female investment[2,3]. Understanding the differences in the immune response between the sexes is a rapidly emerging field with very wide impact. Sex differences in the innate immune response, in particular, have been implicated in outcome from vaccines and infections in adults and children[2,3]. Single stranded RNA viruses such as HIV are sensed by plasmacytoid dendritic cells (pDCs) via expression of TLR7, resulting in higher type I IFN (IFN-I) expression reported in female adults and accelerated activation of antiviral immune responses[4]. TLR7 is expressed on the X chromosome, and there is some degree of TLR7 escape from X chromosome inactivation (XCI)[5]. The stronger IFN-I signalling observed among adult females results in part, therefore, from increased expression of TLR7 on a significant subset of innate immune cells. In HIV, initial control among adults is superior in females[6], who are 5-fold more likely to achieve 'elite control' (suppression of viraemia in the absence of antiretroviral therapy (ART)) than males[7]. However, increased immune activation, which 'fuels the fire' of HIV infection[8], increases immunopathology and HIV disease progression in females in chronic infection[4,6]. In autoimmune diseases such as SLE where TLR7 dosage is a key pathogenic factor[5], affected females outnumber males by 9:1. Likewise, vaccine-specific immune responses are stronger, but adverse events are also more frequent, in females of all ages[2,3].

In the present study we observe increased female susceptibility to in utero HIV infection, and show that this is associated with the preferential transmission of low replication capacity, IFN-I-resistant viruses to females. We then proceed to explore the potential impact of sex-specific innate immune selection of HIV-1 in fetal life on subsequent HIV remission ('functional cure').

## Results
**Increased female susceptibility to in utero HIV transmission**. Consistent with six similar studies[9–11] (Fig. 1a), we observed in a cohort of >170 in utero HIV-infected infants in KwaZulu-Natal, South Africa, that infected females significantly outnumber males by 1.7:1. By contrast, males outnumber females among HIV-unexposed uninfected children and HIV-exposed uninfected children[9]. Overall in KwaZulu-Natal, 50.6% of births are males[12].

To distinguish between the alternative explanations of increased female susceptibility to infection versus increased in utero death in male fetuses following infection, we studied infants born to mothers who themselves became infected during pregnancy (that is, mothers testing HIV-negative initially and then subsequently HIV-positive when re-tested). Our results show that the number of male and female infants who had been exposed to seroconverting mothers was not significantly different, the odds ratio of in utero-infected females in this setting was 2.8 ($p = 0.027$, Fisher's Exact test, Fig. 1b), consistent with the notion of increased female susceptibility.

Further evidence of female susceptibility is suggested by our results showing that the viruses (*Gag-Pro* sequences) transmitted to females in utero had significantly lower replicative capacity than those transmitted to males ($p = 0.0005$, Fig. 1c). Assessment of the replicative capacity of virus from the mothers of infected females indicated that female permissiveness to infection via low replicative capacity virus occurred not because of lower replication capacity of viruses to which they were exposed (Fig. 1d). In vitro experiments, using a laboratory-adapted strain of virus, comparing HIV infection levels in cord blood from 19 sets of

HIV-uninfected, sex-discordant twins, showed preferential infection in CD4 T-cells from females (Fig. 1e), demonstrating that host factors play a role in the observed increased susceptibility to in utero-HIV infection in females.

**Preferential transmission of IFN-I-resistant viruses to females**. A clue to the possible mechanisms underlying this increased female susceptibility is that the sex differences in in utero infection are most marked in the setting of recent maternal infection (Fig. 2a, b, Supplementary Fig. 1). In adults, circulating viruses in chronic infection are predominantly IFNα-sensitive but the viruses transmitted are IFNα-resistant[13,14]. Since females make stronger IFNα responses to HIV and have higher levels of immune activation for a given viral load than males[4], we hypothesised that female fetuses also have higher levels of immune activation and are more susceptible than males to infection via IFNα-resistant viruses; and, in addition, that recently infected mothers are more likely to harbour IFNα-resistant virus than chronically infected mothers. Consistent with this hypothesis, HLA-DR expression on CD4+ T-cells in cord blood in HIV-uninfected sex-discordant twins was indeed significantly higher in females (Fig. 2c), correlating with the levels of CCR5 expression and of HIV-infection in vitro 10 days later (Fig. 2d). Analysis of in utero transmitted viruses in 22 infants at baseline (at a median of 5.5 days of age) showed that viruses transmitted to females are also more IFNα-resistant than viruses transmitted to males (Fig. 2e, f). However, although numbers studied were small, viruses harboured by mothers who seroconverted during pregnancy did not appear more IFNα-resistant than those harboured by chronically infected mothers (Fig. 2g; in these mothers, the first positive HIV test was a median of 3 days versus >6 years, respectively, prior to delivery). Thus, the sex bias within in utero mother-to-child transmission (MTCT) being most marked in the setting of acute maternal infection is not explained here by changes in viral IFNα-sensitivity over the course of infection. However, these data show sex-specific selection of the virus, and that mothers transmitting to females are more likely to carry and transmit IFNα-resistant virus to females, irrespective of how long they have been infected.

**Superior maintenance of aviraemia in ART non-adherent males vs females**. To investigate the impact of immune sex differences in utero and sex-specific adaptation of transmitted virus on outcome in these HIV-infected infants, we first compared baseline plasma viral load in female and male infants. Whereas in adult infection initial plasma viraemia is significantly lower in females compared to males[6], this is not the case following in utero transmission in the current study (Fig. 3a) or in previous studies of in utero-infected infants[15]. Indeed, baseline DNA viral loads are also somewhat higher in female than male infants (Fig. 3b). Since DNA viral load is currently the best predictor of post-treatment control of viraemia[16,17], or 'functional cure' of HIV following ART discontinuation, these data prompt the hypothesis that male infants following in utero-HIV transmission have greater cure potential than do females.

To test whether individuals on ART have achieved functional cure requires ART interruption, since an undetectable DNA viral load does not necessarily imply post-treatment control post-ART[18–21]. Although planned ART interruption was not undertaken in this cohort, it is unfortunately the case that the majority of HIV-infected infants are ART non-adherent, for the reason that ART non-adherence in the mother is strongly associated both with mother-to-child-transmission arising in the first place, and with subsequent ART non-adherence in the child[22,23]. Although there was no sex difference in time to suppression of

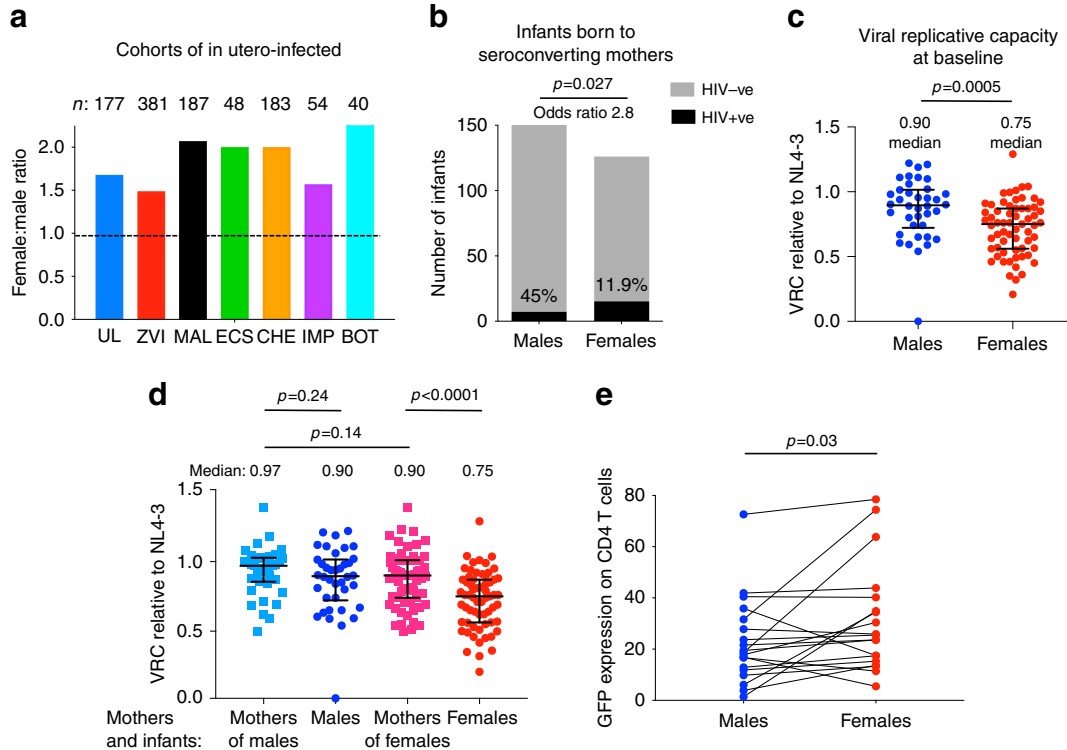

**Fig. 1 Increased female susceptibility to in utero HIV infection compared to males. a** Female:male sex ratio in the current Ucwaningo Lwabantwana cohort of in utero HIV-infected infants and in six other similar cohorts. UL: Ucwaningo Lwabantwana; ZVI: Zvitambo[9]; MAL: Malawi[10]; ECS: European Collaborative Study[11]; CHE: CHER; IMP: IMPAACT P1115; BOT: Botswana. These data refer to in utero infected infants only, except for the CHER study which included intra-partum and in utero infected infants. **b** Numbers of male and female infants exposed to mothers seroconverting during the pregnancy (a negative HIV antibody test followed later in the pregnancy by a positive antibody test) at Queen Nandi Hospital, Empangeni, Stanger Hospital, Stanger, and Mahatma Gandhi Memorial Hospital, between 2016 and 2018, and numbers of in utero infected infants. The numbers of exposed infants did not differ significantly between the sexes but the number of infections did ($p = 0.027$, 2-tailed, Fisher's Exact test). **c, d** Viral replicative capacity at baseline from 101 in utero infected infants, 63 females and 38 males, and mothers. Data are presented showing medians and interquartile ranges. In panel (**c**), the statistical test used was the Mann–Whitney test. In panel (**d**) the statistical tests used were the Wilcoxon matched-pairs signed rank test (comparing males versus mothers of males), the unpaired paired $t$ test (comparing mothers of males versus mothers of females) and the paired $t$ test (comparing females versus mothers of females). In all cases P values were two-tailed. **e** In vitro HIV infection of cord blood CD4+ T-cells from 19 pairs of HIV-uninfected sex-discordant twins. The statistical test used was the Wilcoxon matched-pairs signed rank test (2-tailed). For Fig. 1c–e, source data are provided as a Source Data file.

viraemia (<20 HIV RNA copies/ml, Fig. 3c), maintenance of plasma aviraemia was significantly superior in males (Fig. 3d, e), even among those infants whose DNA viral loads were measured as 0 cpm peripheral blood mononuclear cell (PBMC) (Fig. 3d). Male or female infants born to recently infected mothers were more successful at maintaining aviraemia than infants born to chronically infected mothers (Fig. 3f). A LASSO-regularised Cox regression analysis[23,24] confirmed recent maternal infection, male sex of the in utero-infected infant, and the interaction between these two parameters, as the three principal factors predicting sustained maintenance of aviraemia in in utero-infected children among 10 parameters analysed (Fig. 3g–h). In this analysis of the 119 infants who achieved suppression of viraemia (plasma HIV viral load <20 c/ml), the influence of these parameters on time until viral rebound (plasma HIV viral load >20 c/ml) was analysed, infants who maintained suppression of viraemia (plasma HIV viral load <20 c/ml) being treated as right-censored.

A possible explanation for improved outcome in males might be that mothers of male infants are more adherent than mothers of female children in administering ART to their infants. As stated above, suppression of viraemia on ART is strongly correlated within mother-child pairs (Fig. 3I, j), because of shared ART non-adherence rather than shared ART resistance[25]. Comparing male infants and mothers of males with female

infants and mothers of females, viral suppression is correlated within pairs (Supplementary Fig. 2A) and there is no difference in viral suppression between males and females or between mothers of males and mothers of females (Fig. 3k). By contrast, comparing recently infected mothers and their infants with chronically infected mothers and their infants, viral suppression remains correlated within pairs (Supplementary Fig. 2B) but here there is a significant difference in viral suppression between recently and chronically infected mothers ($p = 0.0004$) and between the infants of recently and chronically infected mothers, respectively ($p = 0.01$, Fig. 3l). These data suggest that chronically infected mothers and their children are more likely to be ART non-adherent than recently infected mothers and their children; but that mothers of females and female children are not more likely to be ART non-adherent than mothers of males and male children.

Consistent with the notion that viral rebound following ART interruption is less likely to occur in males, case report forms record that, in three cases, the caregivers of males who achieved undetectable DNA viral loads gave a history of failing to administer ART for significant periods of time without viraemia subsequently resulting in the child. In one instance, no ART was collected from the pharmacy for 2 months and the child received no therapy for at least 7 weeks; in a second case, none of the prescribed ART medications were detectable in the child's plasma

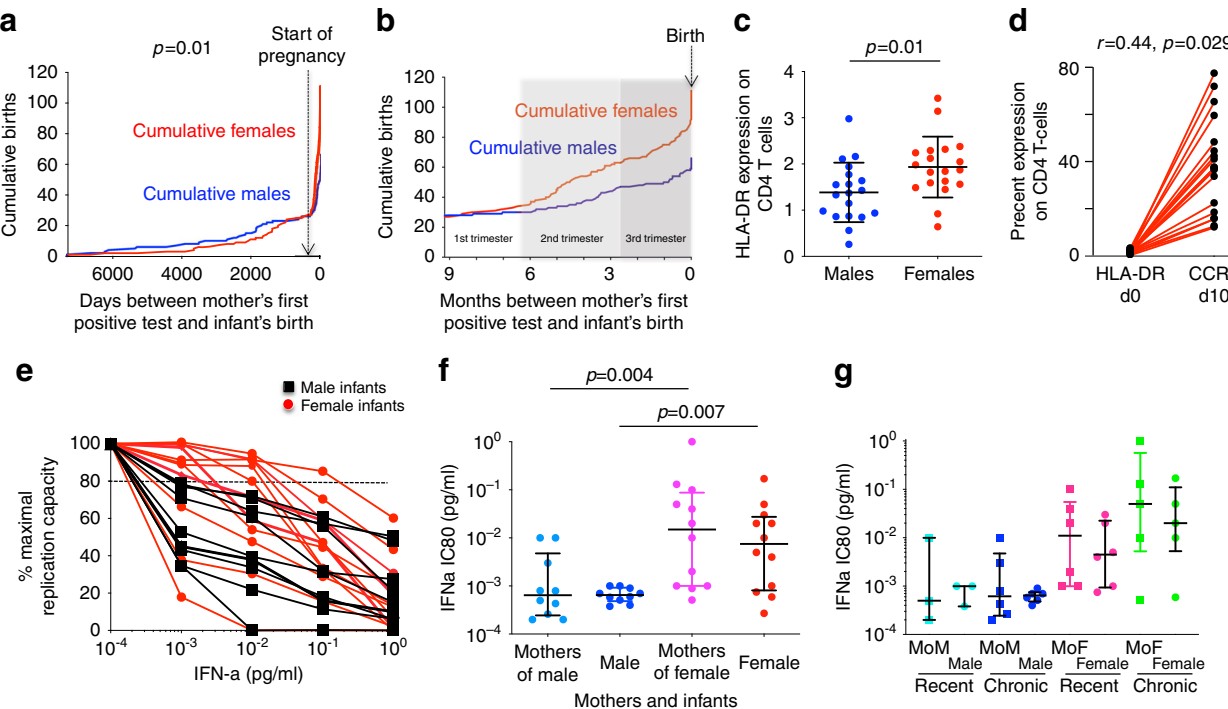

**Fig. 2 Impact of timing of maternal infection in relation to sex differences in intrauterine transmission and in interferon-sensitivity of in utero transmitted viruses. a** Cumulative number of male and female in utero transmissions by time (days) between mother's first HIV-positive test and the birth of the infant. Start of pregnancy shown as 280 days prior to delivery for simplicity (some infants were born prematurely). *P* value denotes sex difference in infants born to mothers whose first HIV test was prior to pregnancy (defined as 'chronically infected') versus those whose mothers' first HIV test was during the pregnancy (defined as 'recently infected') (Fisher's Exact test, 2-sided). **b** Cumulative number of male and female in utero transmissions by time (months) between mother's first HIV-positive test and the birth of the infant in the mothers who first tested positive during the pregnancy. As in (**a**), start of pregnancy shown as 280 days prior to delivery. **c** HLA-DR expression in cord blood CD4+ T-cells in 19 sets of HIV-uninfected sex-discordant twins. Medians and interquartile ranges are shown; the statistical test used was the unpaired *t* test (2-sided). **d** Correlation between CCR5 expression on CD4 T cells on day 10 and HLA-DR expression on CD4 T cells on day 0 in females. The correlation between GFP on day 10 with baseline HLA-DR expression was *r* = 0.35, *p* = 0.07 in females. In males, correlations between HLA-DR expression on day 0 and CCR5 expression on day 10 and GFP expression on day 10 were weaker (*r* = 0.05 and 0.18, respectively; *p* = 0.41 and 0.23, respectively). In all cases, the *t* test used and *p* values were one-sided). **e, f** IFN-α sensitivity of *Gag-Pro*-NL4-3 chimeric viruses derived from baseline virus from 22 infants (10 male, 12 female). Raw data for the 22 infants are shown in panel (**e**) IC80 values and *p* values from these raw data are shown for the 22 infants and mothers of the infants in panel (**f**). The statistical test used in each case was the Mann–Whitney test, *p* values were 2-sided. Medians and interquartile ranges are shown. **g** Data as shown in panels (**e, f**) but also showing timing of maternal infection as categorised above: 'chronic': first HIV test prior to pregnancy; or 'recent': first HIV test during current pregnancy. Medians and interquartile ranges are shown. MoM: mother of male child; MoF: mother of female child. Source data are provided as a Source Data file.

for a period of 7 months; and, in the third, the parent reported missing at least 4 doses a week over a 6 month period and only 1 of the 4 prescribed ART medications was detectable at therapeutic levels when plasma was analysed (Fig. 4, Supplementary Fig. 3A). Among the 17 children who achieved undetectable DNA viral loads, there was no difference between males and females in the frequency of ART non-adherence, as documented on the case report form at each clinic visit. However, males maintained aviraemia significantly better than females in children whose caregivers gave a history of ART non-adherence (*p* = 0.029, Supplementary Fig. 3B).

## Discussion

These data, in showing sex-specific innate immune selection of HIV associated with an increased female susceptibility to in utero-HIV infection, demonstrate that immune sex differences exist prenatally and matter. In Hepatitis C virus (HCV) infection, female infants are also twice as susceptible to mother-to-child transmission[26,27]. In congenital cytomegalovirus (CMV), females have 3-fold more neurological disease[28]. In addition to the increased risk of in utero-HIV infection among females, we also

report here sex differences in outcome, with enhanced cure potential among males.

The finding here that baseline plasma RNA and proviral DNA loads are somewhat lower in in utero-infected males is consistent with previous studies showing higher plasma viral loads in females in the first year of life and lower viral loads than males in later childhood[15,29]. In adults and in later childhood, a broad, high-frequency HIV-specific CD8+ T-cell response plays a central role in control of HIV, whereas, in the first 1–2 years of life, HIV-specific CD8+ T-cell activity is narrowly based, low-frequency and therefore relatively ineffective[30]. The advantages of a stronger IFN-I response to HIV in females is therefore maximal in adults and older children, in whom elite control is 5-fold and 10-fold more frequent, respectively, in females than males[6,31]. In utero, however, the higher level of immune activation observed even in HIV-unexposed females would be expected to increase CCR5 expression[32] and provide more targets for HIV infection. Whereas in adult transmission studies, transmission of lower replicative capacity viruses results in lower proviral DNA loads and improved outcome in the recipient[33], this does not appear to be the case following in utero-transmission of low replication capacity viruses to females. The fact that these low

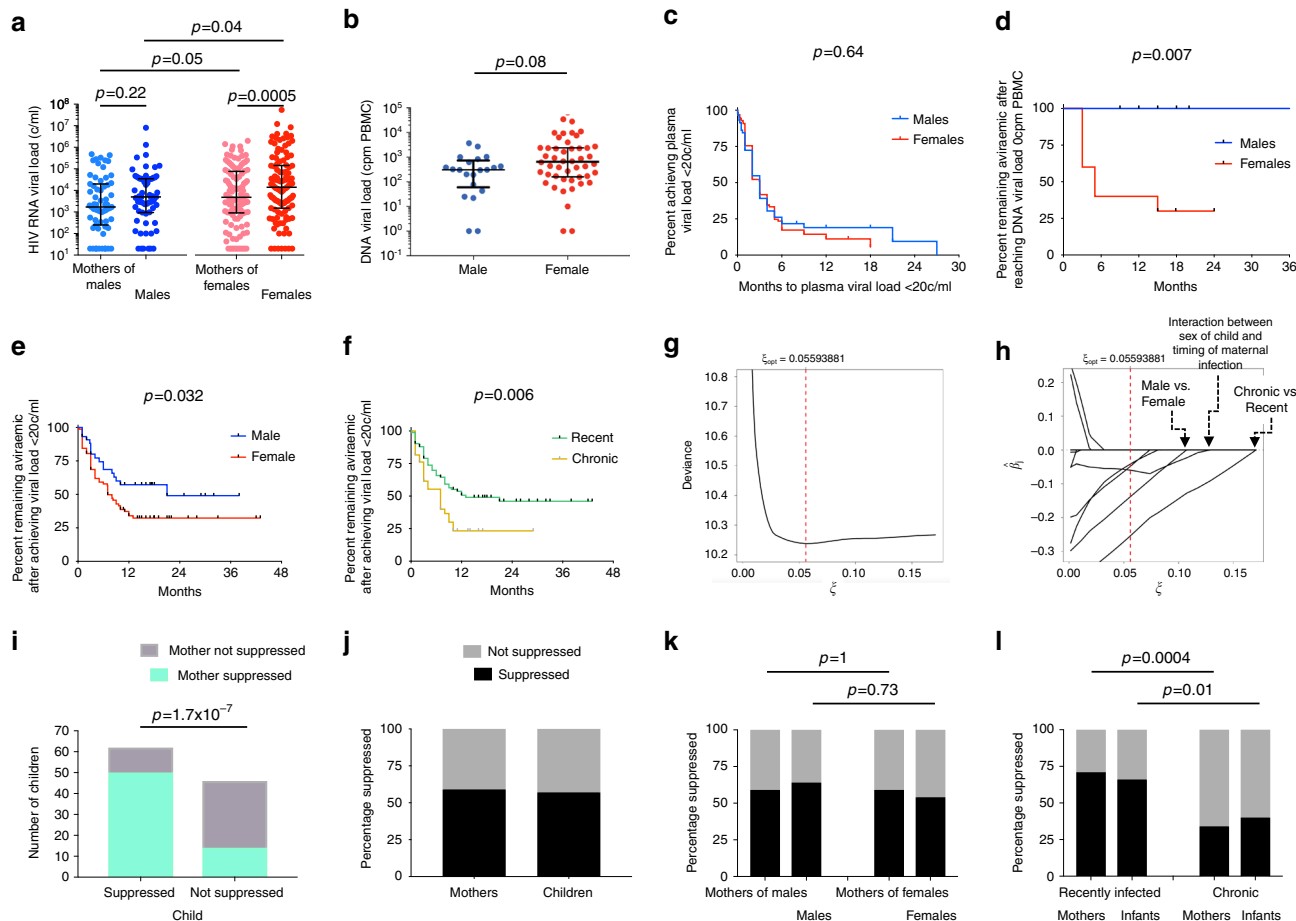

**Fig. 3 Impact of sex differences and timing of maternal infection on outcome post-infection on plasma RNA and DNA viral loads, and viral rebound on ART. a** Baseline plasma RNA viral loads in 177 mother-child pairs. The statistical tests used were the Mann–Whitney test (comparing males versus females and comparing mothers of males versus mothers of females) and the Wilcoxon matched-pairs signed rank test (comparing males versus mothers of males and comparing females versus mothers of females). In each case, p values were 2-sided. Medians and interquartile ranges are shown. **b** Baseline total DNA viral loads in 69 in utero-infected infants (48 female, 21 male). The statistical test used was the Mann–Whitney test, the p value was 2-sided. **c–f** Kaplan–Meier curves showing time to achieve plasma HIV RNA load of <20 copies/ml in males and females (panel **c**), time to viral rebound (>20 copies/ml) in males and females (panels **d**, **e**), time to viral rebound (>20 copies/ml) in in utero-infected children born to recently and chronically infected mothers (panel **f**). In panel D, data are shown from the 17 subjects who reached DNA viral loads reported as 0 cpm PBMC. In each case, the statistical tests used were the Log-rank (Mantel–Cox) test. **g** The cross-validation error for a grid of LASSO penalty parameters $\xi$ in order to determine the optimal amount of penalisation. **h** The coefficient paths of the (standardised) regression parameter estimates versus the LASSO penalty parameter $\xi$; the optimal $\xi$ is indicated by the vertical red dashed line. Re-standardised coefficients of the coefficients selected by the model were: Recent maternal infection: −0.552; Interaction between infant sex and timing of maternal infection: −0.146; Male sex of infant: −0.101; Age of infant at baseline: −0.023; plasma RNA viral load at baseline: $7.4 \times 10^{-9}$. Parameters not selected were: infant baseline absolute CD4 count, CD4%, absolute CD8 count, CD8%, and CD4:CD8 ratio. **i–l**. Suppression of viraemia (plasma viral load <20 copies/ml) in mother-child pairs using most recent timepoint studied. Panels **i–j**: all mother child pairs studied (n = 108); panel **k**: mothers of males and males (39 pairs) compared with mothers of females and females (69 pairs); panel **l**: recently infected mothers and children (73 pairs) compared with chronically infected mothers and children (35 pairs). In panels **i–l**, the statistical tests used was Fisher's Exact test; p values were 2-sided. Source data are provided as a Source Data file.

replication capacity viruses are also relatively IFNα-resistant may contribute to the higher viral loads observed initially in infected females. The consequence of these effects is a viral reservoir in males that is not only smaller than in females but also one composed of IFNα-sensitive virus. In the setting of ART interruption, when initially rebounding virus is strongly IFNα-resistant, the IFNα-sensitive virus emerging in males from previously latently infected cells is potentially more subject to immune control. With this in mind, the recent case of a male South African child[34] who has maintained functional cure for approaching a decade is noteworthy, contrasting with the female Mississippi child[18] who eventually rebounded. Although these are anecdotal cases, they are consistent with the greater potential for functional cure among males proposed here. Further studies

involving planned analytical treatment interruption, as opposed to unplanned ART non-adherence, will be necessary to evaluate more fully the impact of sex differences on functional cure potential.

The observation of sex differences in IFNα-sensitivity in viruses transmitted in utero to males and females, respectively, implicates the TLR7-mediated response of pDCs to HIV in this process, as has been well reported in adult HIV infection[4]. The mechanisms underlying sex differences in response to TLR ligands such as HIV include TLR7 escape from XCI as described above[5], as well as epigenetic and hormonal factors. Although, in utero, males and females are both exposed to high levels of maternal and placental estrogens and progesterone, a clear sex difference in sex hormone levels in utero is testosterone produced

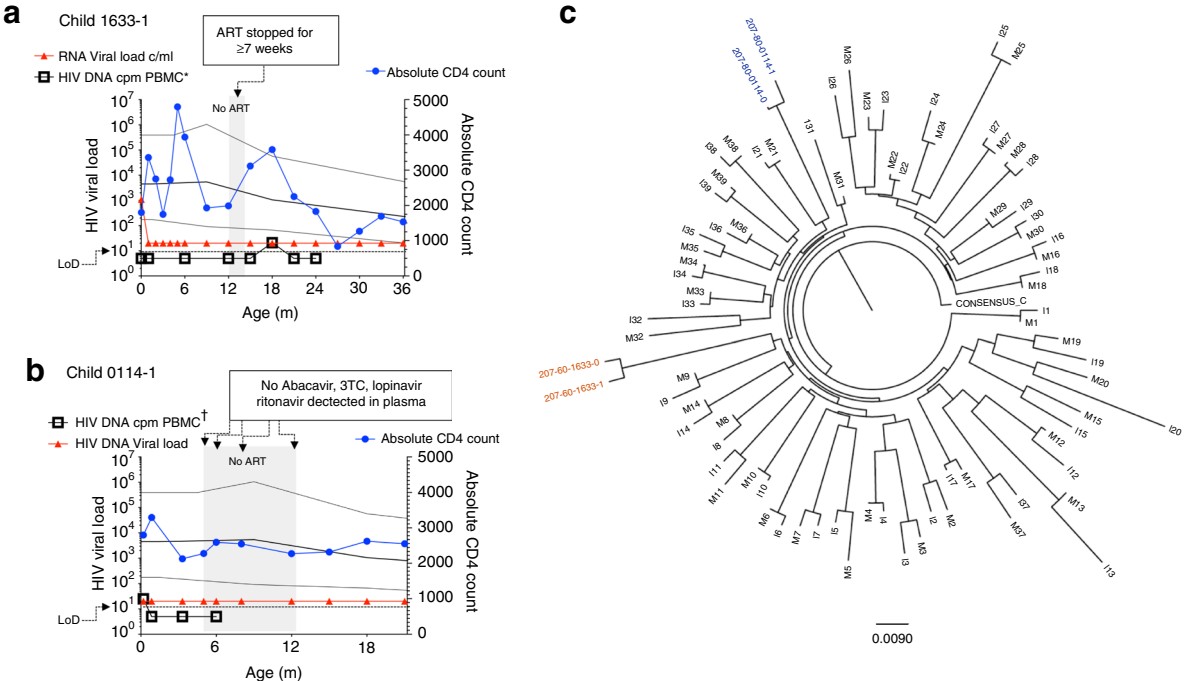

**Fig. 4 Lack of viral rebound in two male subjects following significant periods of ART non-adherence. a** Male subject 1633-1 received no ART for at least 7 weeks at 12–14 months' age: pharmacy records show that the aunt of the child did not collect any ART medication for 2 months following the return of the child's mother to work 12 m after the child's birth. ART was restarted 7 days prior to the viral load test at 15 m. Undetectable DNA viral loads are shown as 5 HIV DNA cpm PBMC. Only the 18 m timepoint was above the limit of detection (DNA load 21 cpm PBMC). **b** Male subject 0114-1: none of the drugs prescribed were detected in plasma tested over a 7-month period (age 5–12 months). At age 15 months and 18 months, only 1 and 2 of the 4 drugs prescribed were detected at therapeutic levels, respectively. Undetectable DNA viral loads are shown as 5 HIV DNA cpm PBMC. Only the baseline timepoint (at 7 days' age) was above the limit of detection (DNA load 25 cpm PBMC). **c** *Gag* was amplified from plasma from the baseline sample from both these mother-child pairs and the sequences determined clustered on a phylogenetic tree, consistent with HIV infection and mother-to-child transmission in each case. Diagnosis in these cases, as in all the in utero-infected infants in the cohort, was determined by 2 or more tests from separate blood samples detecting HIV nucleic acids.

by males from 8–14 weeks' gestation[35]. Sex hormones influence the TLR7-interferon signalling pathway[4,36], testosterone being highly anti-inflammatory[37]. The higher level of testosterone in males in utero may also explain the fact that DNA methylation of autosomal CpG sites is 5-6-fold more frequent in males, most (74%) sex-specific CpG methylations arising prenatally[38].

The stronger IFN-I response observed in adult females might, at first sight, be expected to decrease, rather than increase susceptibility to HIV transmission. However, studies in the SIV macaque model illustrate the complex effects of manipulations to enhance IFN-I signalling. Administration of IFNα initially prevents systemic infection, but continued IFNα treatment enables infection with an increased reservoir size[39]. The data presented here show that females are more susceptible to IFN-I-resistant virus, but that males are more susceptible to IFN-I-sensitive virus (Fig. 2f). Thus, in a setting where mothers are recently infected and the majority of viral quasispecies in acute infection are IFN-I-resistant, as suggested by some[13,14], but not other[40], adult transmission studies, females would be especially susceptible to in utero MTCT, as observed here (Fig. 2a, b). Further studies will be needed to evaluate whether a high frequency of IFN-I-resistant virus in acute adult infection plays a part in increased female susceptibility to in utero MTCT in the setting of maternal infection during pregnancy.

The findings of sex differences in viruses transmitted via mother-to-child prompts the question of whether similar differences arise in adult-to-adult transmission. Sex-specific adaptation of HIV has not been described, but in HCV infection, in which the IFN-I response is strongly implicated in clearance of infection,

host IFN-λ4 expression selects viral polymorphisms that modulate viral load[41]. The observation here that females are more permissive to in utero HIV infection via low replicative capacity viruses is exactly as described in an adult transmission study in which replicative capacity was inferred by viral sequence analysis[42]. These findings together suggest that the vulnerability of female adolescents and young women in sub-Saharan Africa to HIV infection[43,44], where infected females outnumber males by 2–3:1, has an important biological component. Our findings further indicate that, at the same time that socio-economic and cultural factors contributing to the epidemic are addressed, it is important to take into account increased female susceptibility to HIV infection in the development of effective prevention strategies.

The observation of sex-specific innate immune selection of HIV described here is an example of a phenomenon that is well described in the animal kingdom in which infection levels, disease and virulence differ between the host sexes for a range of pathogens[45]. The impact of sex-specific adaptation in settings where those pathogens are also transmitted via mother-to-child is less well-studied. Here we describe in utero MTCT, preferentially to females, of IFN-I-resistant HIV that evades the stronger IFN-I response that has been shown in adult females. It is possible that sex-specific innate immune selection of viruses may influence infection susceptibility and disease outcome in other scenarios, such as the MSM (men who have sex with men) HIV epidemic, or in other infections, such as HCV. However, outside of the mother-to-child transmission setting, the multiple additional factors contributing to adult infections and disease will make this concept of pathogenesis more challenging to investigate.

## Methods

**Study subjects.** The Ucwaningo Lwabantwana (meaning 'Learning from Children') is a cohort of 177 in utero infected children enroled in KwaZulu-Natal, South Africa from 2015 to 2019. All infants received antiretroviral therapy (ART) in the delivery room within minutes of birth according to local guidelines. Infants of mothers at high risk of in utero HIV transmission ($n = 81$) were tested for HIV-1 as soon as possible after birth using point-of-care (PoC) testing to detect total nucleic acid (TNA) PCR on whole blood (GeneXpert Qualitative HIV-1 PCR, Cepheid, Sunnyvale, CA, USA) or by overnight plasma HIV RNA viral load measurement (Nuclisens EasyQ v2·0 HIV-1 RNA PCR, bioMérieux, Marcy l'Etoile, France). Infants with a positive or indeterminate result from the standard-of-care (SoC) laboratory based dried blood spot TNA PCR (COBAS AmpliPrep /COBAS TaqMan HIV-1 Qualitative PCR v2, Roche Molecular Diagnostics, Basel, Switzerland) at birth were also enroled ($n = 96$). In all 177 enrollees, confirmed diagnosis required two separate tests detecting HIV nucleic acid. Baseline data were collected at a median of 1.0 (IQR 0.9–1.8) day of age and 11 (IQR 9–14) days of age from the PoC-diagnosed and SoC-diagnosed infants, respectively. Initial combination ART (cART) for infants with confirmed HIV infection comprised twice daily NVP, AZT and lamivudine (3TC) as per local guidelines. This regimen was switched to ritonavir-boosted lopinavir (LPVr), 3TC and abacavir (ABC) at 42 weeks corrected gestational age or at 1 month of age. Mother and infant follow-up occurred monthly for 6 months then 3-monthly. At each visit, blood was drawn for CD4+ T cell quantification, plasma viral load (HIV-1 RNA PCR, NucliSens), and storage of peripheral blood mononuclear cell (PBMC) and plasma. This study was approved by the Biomedical Research Ethics Committee of the University of KwaZulu-Natal and the Oxfordshire Research Ethics Committee. Written informed consent for the infant and mother's participation in the study was obtained from the mother or infant's legal guardian.

We assessed viral infectivity in cord blood PBMC from 19 pairs of sex discordant twins from Kimberley Hospital, Northern Cape, South Africa. Informed consent was obtained from all participating subjects. Studies were approved by the University of the Free State Ethics Committee, Bloemfontein and the Research Ethics Committee, University of Oxford.

**Viral RNA isolation and nested RT-PCR amplification of *gag-protease* from plasma.** Viral RNA was isolated from plasma by use of a QIAamp Viral RNA Mini Kit from Qiagen. The Gag-Protease region was amplified by reverse transcription (RT)-PCR from plasma HIV-1 RNA using Superscript III One-Step Reverse Transcriptase kit (Invitrogen) and the following Gag-protease-specific primers: 2cRx; 5′ CAC TGC TTA AGC CTC AAT AAA GCT TGC C 3′ (HXB2 nucleotides 512 to 539) and 623Fi; 5′ TTT AAC CCT GCT GGG TGT GGT ATT CCT 3′ (nucleotides 2851 to 2825). Second round PCR was performed using 100-mer primers that completely matched the pNL4-3 sequence using Takara EX Taq DNA polymerase, Hot Start version (Takara Bio Inc., Shiga, Japan). One hundred microliters of reaction mixture was composed of 10ul of 10x EX Taq buffer, 4ul of deoxynucleoside triphosphate mix (2.5 mM each), 6ul of 10uM forward primer, Gag-Pro F; (GAC TCG GCT TGC TGA AGC GCG CAC GGC AAG AGG CGA GGG GCG GCG ACT GGT GAG TAC GCC AAA AAT TTT GAC TAG CGG AGG CTA GAA GGA GAG AGA TGG G, 695 to 794) and reverse primer, Gag-Pro R; (GGC CCA ATT TTT GAA ATT TTT CCT TCC TTT TCC ATT TCT GTA CAA ATT TCT ACT AAT GCT TTT ATT TTT TCT TCT GTC AAT GGC CAT TGT TTA ACT TTT G, 2646 to 2547), 0.5ul of enzyme, and 2ul of first round PCR product and DNase-RNase-free water. Thermal cycler conditions were as follows: 94 °C for 2 min, followed by 40 cycles of 94 °C for 30 s, 60 °C for 30 s, and 72 °C for 2 min and then followed by 7 min at 72 °C. PCR products were purified using a QIAquick PCR purification kit (Qiagen, UK) according to manufacturer's instructions.

**Generation of recombinant *Gag-Protease* viruses.** A deleted version of pNL4-3 was constructed that lacks the entire Gag and Protease region (Stratagene Quick-Change XL kit) replacing this region with a BstE II (New England Biolabs) restriction site at the 5′ end of Gag and the 3′ end of protease. To generate recombinant viruses, 10 μg of BstEII-linearised plasmid plus 50 μl of the second-round amplicon (approximately 2.5 μg) were mixed with $2 \times 10^6$ cells of a Tat-driven green fluorescent protein (GFP) reporter T cell line (GXR 25 cells) in 800 μl of R10 medium (RPMI 1640 medium containing 10% fetal calf serum, 2 mM L-glutamine, 100 units/ml penicillin and 100ug/ml streptomycin) and transfected by electroporation using a Bio-Rad GenePulser II instrument (300 V and 500 uF). Following transfection, cells were rested for 45 min at room temperature, transferred to T25 culture flasks in 5 ml warm R10 and fed with 5 ml R10 on day 4. GFP expression was monitored by flow cytometry (FACS Calibur; BD Biosciences), and once GFP expression reached >30% among viable cells supernatants containing the recombinant viruses were harvested and aliquots stored at −80 °C.

**Viral replication capacity assays.** The replication capacity of each chimera is determined by infection of GXR cells at a low multiplicity of infection (MOI) of 0.003. The mean slope of exponential growth from day 2 to day 7 was calculated using the semi log method in Excel. This was divided by the slope of growth of the wild-type NL4-3 control included in each assay to generate a normalised measure of replication capacity. Replication assays were performed in triplicate, and results were averaged. These VRC determinations were undertaken entirely blinded to the identity of the study subject.

**Viral sequencing and phylogenetic analysis.** Population sequencing was undertaken using the Big Dye ready reaction terminator mix (V3) (Department of Zoology, University of Oxford) using *gag-protease* sequencing primers SQ2FC (CTT CAG ACA GGA ACA GAG GA), GF100-1817.18 (TAG AAG AAA TGA TGA CAG), gf2331 (GGA GCA GAT GAT ACA GTA TT), SQ16RC (CTT GTC TAG GGC TTC CTT GGT), GAS4R (GGT TCT CTC ATC TGG CCT GG), pan1dRx (CAA CAA GGT TTC TGT CAT CC), GR1981 (CCT TGC CAC AGT TGA AAC ATT T), and gr2536 (CAG CCA AGC TGA GTC AA). Sequence data were analysed using Sequencher v.4.8 (Gene Codes Corporation). Nucleotides for each gene were aligned manually in Se-Al v.2.0a11. Maximum-Likelihood phylogenetic trees were generated using PHYm131[46] (http://www.hiv.lanl.gov) and visualised using Figtree[47] v.1.2.2 (http://tree.bio.ed.ac.uk/software/figtree/).

**IFN Resistance assays.** To determine IFNα2a concentrations required to inhibit virus replication to 80% (IC80), GFP reporter T cells (GXR 25 cells) were left untreated or cultured in the presence of increasing amounts of IFNα2a (0.00074–5.5 pg/ml), infected with equal amounts of patient specific *gag-pro* chimeric virus (MOI 0.03) and cultured for 7 days. IFN-containing medium was replenished every 24 h. Viral replication was measured by GFP expression and the mean slope of exponential growth from day 2 to day 7 was calculated using the semi log method in Excel. Viral replication was plotted for each IFN concentration as the percentage of viral growth in the absence of IFN. Replication in the untreated cells was used to determine the replicative capacity of *gag-pro* viral isolates.

**In vitro HIV infectivity assays in sex-discordant twins.** Cord blood mono-nuclear cells (CBMC) were cryopreserved from cord blood samples of sex-discordant twins recruited at Kimberley Hospital, South Africa. PBMC's were thawed in RPMI with 20% fetal calf serum medium, rested for 1 h at 37 °C, 5% $CO_2$; and 1 million stained in a total volume of 50ul staining solution with titrated concentrations of fluorochrome conjugated monoclonal antibodies against cell surface markers (CD3, CD4, CD8, PD-1, HLA-DR, CCR5, CD45RA, CCR7) and Invitrogen near-IR Live/dead marker for 30 minutes at RT in the dark. Cells were then washed twice in PBS and fixed in 2% paraformaldehyde. Samples were acquired on an LSR II (BD) flow cytometer within 12 h of staining and analysed using FlowJo version 8.8.6. The remaining PBMC were activated with PHA for 72 h then infected with an R5 tropic HIV-GFP tagged virus at MOI 0.01 and cultured for 7 days. R10 medium was replenished every 24 h. Viral infectivity was measured as the amount of CD4+ T cells expressing GFP at day 7 post-infection. The R5 tropic HIV-GFP tagged virus was produced by transfecting 293 T cells with the plasmid construct R5-HIV-1-GFP[48] by CalPhos (Invitrogen, Spain)[49]. After 48 h, viral supernatant was harvested and stored at −80 C, and the 50% tissue culture infective dose ($TCID_{50}$) was determined on TZM-bl cells (NIH AIDS Reagent Program, USA) using the Reed and Muench method[48].

**Plasma HIV RNA viral load and HIV DNA viral load measurement.** Plasma HIV-1 RNA viral load measurement was undertaken using the BioMérieux NucliSens Version 2.0 Easy Q/Easy Mag (NucliSens v2.0) assay (dynamic range 20–10 m). Total HIV-1 DNA levels were quantified using droplet digital PCR (BioRad, Hercules, CA, USA) from PBMC. Samples from all individuals were screened with two primers/probe sets, annealing to the 5′LTR and gag conserved regions of HIV-1 genome. In measurement of viral DNA via ddPCR, limit of detection varied according to sample quality and therefore input cell number. The median limit of detection from the entire dataset quantified was 8 HIV DNA copies/million PBMC (IQR 4-15). In the analyses of subjects with undetectable DNA loads (Fig. 3d) we analysed subjects whose DNA viral load were reported as zero ($n = 17$, $p = 0.008$ comparing viral rebound in males and females). We excluded four subjects whose DNA viral loads were 4, 4, 7 and 8 DNA cpm PBMC, respectively, where in each case the limit of detection for these assays was lower than the measured values, i.e the DNA loads were detectable. However, including these four individuals (3 female, 1 male) in the survival analysis shown in Fig. 3d did not affect the significance of the result ($p = 0.008$ including these four subjects). In Fig. 4 and Supplementary Fig. 3, DNA viral loads reported as 0 cpm PBMC are shown as 5 cpm PBMC on the $\log_{10}$ y-axis.

**Statistical analysis.** In scatterplots, median values and IQRs are indicated. Comparisons were performed using Fisher's exact tests for categorical variables and Student's t test or Mann–Whitney U test for continuous variables. Maintenance of viral suppression was calculated using Kaplan–Meier analysis and different groups were compared using the Log-Rank test. The influence of several predictors on time to plasma viral load >20 HIV RNA c/ml was analysed using a Cox regression model, where the set of relevant predictors was selected by the LASSO penalty approach[23] and the optimal penalty parameter $\xi$ was determined via 10-fold cross validation using the glmnet R package[24]. All p-values were two-sided. All calculations and graphs were performed using R Software and GraphPad Prism v7.

**Reporting summary**. Further information on research design is available in the Nature Research Reporting Summary linked to this article.

## Data availability
The source data underlying Figs. 1–3 and Supplementary Figs. 1–3 are provided as Source Data files. Sequence data from our study were deposited into NCBI Genbank with accession numbers MN957798–MN957801. All other data are available from the corresponding author upon reasonable request.

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

## Acknowledgements
The authors thank Tommaso Pizzari for helpful discussions and comments in reviewing the manuscript. We also wish to thank the mother-child pairs who have participated in this study, together with Noxolo Mchunu, Gugu Hlophe, Sjabulile Ngcobo, Nomfundo Biyela, Nicky Linda, Thandi Sikhakhane, Patience Mthetwa, Nomvula Nzuza, Gugu Mbuyazi and Pamela Small, the study nurses and counsellors working on this project. This work is funded by grants from the National Institutes of Health (RO1AI133673 P.G.), the Wellcome Trust (WT 104748MA, P.G., WT 110110, P.C.M.). JGP is supported by the National Health Institute Carlos III grant PI17/00168 and Redes Temáticas de Investigación en SIDA (ISCIII RETIC RD16/0025/0041); Acción Estratégica en Salud; Plan Nacional de Investigación Científica, Desarrollo e Innovación Tecnológica

2008–2011; Instituto de Salud Carlos III; Fondos FEDER. MCL is supported by the Government of Catalonia's Secretariat for Universities and Research of the Ministry of Economy and Knowledge.

## Author contributions

E.A., J.M., and P.G. wrote the paper, contributed to the study conception and design, and contributed to the acquisition, analysis and interpretation of the data; C.O., J.K., and P.M. contributed to the study design, and to the analysis and interpretation of the data; MAlt contributed to the study conception and design, and to the interpretation of the data; J.M.P. and J.P. contributed to the study design, and to the acquisition, analysis and interpretation of the data; PJ contributed to the study design and to the acquisition of data; N.B., M.M., R.F., K.S., V.N., J.R., V.V., K.G., J.A., N.N., L.M., JvL, Y.G., K.C., C.K., R.B., M.K., R.P.P., M.C.L., M.C.P., N.M., MArc, T.N., and A.G. all contributed to the acquisition and interpretation of the data.

## Competing interests

The authors declare no competing interests.
