## [Peer Review File · Nature Communications]

Reviewers' Comments:

Reviewer #1:

Remarks to the Author:

The primary focus of this research was to identify the causes of increased susceptibility of female fetuses to HIV-1 infection in utero. Through a rigorous series of experiments, the authors show that female fetuses have more IFN-resistant HIV-1 and greater immune activation than male fetuses. Males, on the other hand are better able to maintain low viremia, which is not caused by better adherence of ART or differential levels of infection or chronicity of infection among mothers of males. The systematic and logical experiments conducted clearly addressed alternative explanations, especially pertaining to the role of gender bias. I only have minor modifications and suggestions.

1. Line 70 and elsewhere in the manuscript, the authors should be vigilant about clarifying when a statement pertains to what is known in adults only. All too often statements e.g., 'higher IFN expression in females' occur, when the truth is that 'higher IFN expression in ADULT females' is what is shown in the literature.
2. Line 131. Why do the authors refer to the greater DNA viral load in females as 'somewhat' when the p is less than 0.05. Also, there is inconsistent use of p = ns vs giving an actual p-value for a nonsignificant difference. Be consistent.
3. The most problematic and at times confusing parts of the paper pertained the analysis and interpretation of viral sensitivity to IFN α (basically data in Fig 2E). There is no indication of whether the sex difference was significant when in fact there does not appear to be a difference between viruses isolated from males and females. In the Results, lines 114-124 presents a 'just so' story that makes it sound as though sex-specific adaptation of the virus is more robust and clearer than it actually is.
4. Line 145-46 is just describing data shown in Fig 3F, not Fig 3F-H. On the Y-axis of Fig 3F, indicate that this is the 'percent MALES remaining aviremic'.

Reviewer #2:

Remarks to the Author:

The authors address some important questions pertaining to sex-specific susceptibility in acquisition of HIV-1 infection in utero (IU) through study of more than 170 HIV-1 infected infants from KwaZulu Natal in South Africa – providing some long-awaited answers. This study has been carefully conducted and the manuscript is very well written - and systematically tackles the question of whether in the first place females are more susceptible to IU infection than males, and then goes on to describe features that might explain this outcome as well as that of better maintained virological control that was found in males compared to females. These findings are very important for HIV cure strategies, given that multiple studies/clinical trials are addressing the role of provision of very early ART to IU infants in the context of achieving functional cure in children (although already evident that very early ART alone is not likely to be enough but is an important first step). Importantly, as the main focus of these studies is on IU-infected children, we need a better understanding of the host and viral characteristics of these children - which needs to address both susceptibility to infection and subsequent control of viraemia on ART (with non-adherence providing the test scenario akin to planned ART interruption/pause).

In summary, this study found the following: (i) Increased female susceptibility to IU infection, supported by data showing this was not due to increased male deaths IU (an argument previously raised) and further supported by findings of increased female:male ratios across several cohorts from different locations (note that many of these studies however, may have biases in patient selection as are not classical transmission cohorts); (ii) Females were infected with viruses of lower replicative

capacity; (iii) Cord blood cells from females were more susceptible to HIV-1 infection in vitro (evaluated using cells from sex-discordant uninfected twins), (iv) Higher levels of immune activation in HIV-unexposed females in utero (evaluated using the same HIV-uninfected sex-discordant twins) which also correlated with more permissive HIV-1 infection in vitro; (v) Greater IFN-resistance of viruses transmitted to females; (vi) Males had lower baseline viral loads and more effective maintenance of viral suppression; and (vii) Male infants born to mothers who seroconverted during pregnancy were the group most likely to achieve aviraemia. The data as a collective convincingly show sex-specific adaptation of HIV – associated with female susceptibility to IU infection and suggest infected males may have a greater potential for attaining functional cure.

Specific comments:

Given the justifications for greater susceptibility to IU infection in females – could the same arguments not hold true for intrapartum or breast-feeding infection? Are there any data in the literature that might support this? It seems likely that the features (viral, immune), or at least some, described for susceptibility for IU infection females in this study are unlikely to be different later on, unless immune maturation events substantially change some of these parameters. This could be discussed.

Among children with a history of ART non-adherence, 3 males maintained viral suppression while the 4 females showed viral rebound. The DNA viral loads are very low (mostly <10 copies/million PBMCs; also considering baseline levels which start higher) in the males, if compared for example to the spread of baseline levels of the 21 males and 48 females in Fig 3B - they would be at the lower end of the distribution (some females also have similarly low levels). With DNA VLs serving as a marker for time to rebound, this begs the question of what the trajectories of DNA VLs were in the 4 females who rebounded. Similar graphs could be provided to view the RNA VL trajectories, DNA VLs and CD4 counts of the 4 female cases (in Supplementary), as presented for the male subjects. This would strengthen the findings if DNA VLs were also very low as found in the males. Alternatively, it might be that these particular female cases had much higher levels of HIV DNA, so increasing the probability of rebound (greater viral reservoir to begin with). This could add to further understanding underlying mechanisms involved in the different outcomes of these male and female cases.

With the greater potential for functional cure suggested among HIV-1 infected males, it is interesting that the only case of a male child reported with durable post-treatment control (now for >10 years; infected either IU or IP) has not been mentioned (Violari et al, Nat Comm 2019). This case also had DNA VLs reported as 5 copies/million PBMCs at ART interruption which was the same when tested 9 years later. So DNA VLs are very much in the same range as the male cases highlighted in the current study. This would support to the arguments being made favouring males in post-treatment control.

Small edits:

Reference 5 is incomplete.

Page 25, line 615 – “significant” should be “significance”.

Consistency – spelling of “aviraemic”, Fig3D for example has as “aviremic”.

Figure 2 E-F, G legend: “MoM” etc designations are used only for G, is currently stated under E-F.

Supplementary Figure 3 is also labelled as Suppl Fig 2.

Reviewer #3:

Remarks to the Author:

Adland and colleagues present an interesting and thorough investigation into the differences in perinatal HIV acquisition in male and female infants. They convincingly demonstrate differences in viral acquisition, replication, and suppression in male versus female infants. My enthusiasm for this

work is somewhat diminished by the lack of a consistent theoretical framework for interpreting the results.

Evidence of viral adaptation leading to these differences (as suggested by the Title) is weak. If anything, the TLR7 argument put forth by the authors suggests that viral adaptation is not at play.

I am confused by the framing of this manuscript with regards to immune related sexual selection. Theory would suggest that women are less susceptible to infection and have improved outcomes at the expense of auto-immune outcomes. The findings presented here do not support this hypothesis.

The discussion of these results with regards to "functional cure" are premature (Page 5, line 137). There is no evidence that immediate treatment of perinatally infected infants leads to a functional cure (e.g., the Mississippi baby). Also, see the Abstract, Page 2, line 61.

I am dubious of the relevance of the competition experiments with NL4-3, a subtype B lab adapted strain from the 1980s. I know that these direct competition experiments are not rare in the field. However, competing NL4-3 against subtype C (presumably based on Fig. 4C) virus is of limited value.

Elite Controllers (as first referenced on Page 3, line 73) are often a result of deficient virus, rather than primarily driven by host factors (see PMIDs: 15000695 & 20504921).

Minor Comments:

Figure 4C. This tree needs a scale bar indicating substitutions/site. The nucleotide substitution model implemented in PhyML should be provided. Also, why are there negative branch lengths? The root length obscures the tree structure. Please adjust in FigTree. Finally, both PhyML and FigTree papers need to be cited, rather than just including URLs.

Page 6, line 146. I believe the authors mean "maintaining aviremia".

Reviewer #1:

1. Line 70 and elsewhere in the manuscript, the authors should be vigilant about clarifying when a statement pertains to what is known in adults only. All too often statements e.g., 'higher IFN expression in females' occur, when the truth is that 'higher IFN expression in ADULT females' is what is shown in the literature.

Response: We have reviewed the manuscript and have clarified in line 70 and in the other instances where we have not specified whether adults or children or both are being referred to.

2. Line 131. Why do the authors refer to the greater DNA viral load in females as 'somewhat' when the p is less than 0.05. Also, there is inconsistent use of p = ns vs giving an actual p-value for a nonsignificant difference. Be consistent.

Response: The 'somewhat' referred to the size of difference in DNA load as well as the p value which is 0.08 (Fig 3B). We have in Fig 3C, 3K and Suppl Fig 3B replaced p=ns with the actual p value as suggested.

3. The most problematic and at times confusing parts of the paper pertained the analysis and interpretation of viral sensitivity to IFN α (basically data in Fig 2E). There is no indication of whether the sex difference was significant when in fact there does not appear to be a difference between viruses isolated from males and females. In the Results, lines 114-124 presents a 'just so' story that makes it sound as though sex-specific adaptation of the virus is more robust and clearer than it actually is.

Response: The raw data are shown for all the paediatric subjects studied in Fig 2E. The IC80 values obtained from these data are shown in Fig 2F and the p values are also shown in Fig 2F. In addition in Fig 2F are shown the IC80s for the mothers of male and female infants.

To clarify we have added the following to the Figure legend (lines 534-535):

“Raw data for the 22 infants are shown in panel E. IC80 values and p values from these raw data are shown for the 22 infants and mothers of the infants in panel F.”

4. Line 145-46 is just describing data shown in Fig 3F, not Fig 3F-H. On the Y-axis of Fig 3F, indicate that this is the ‘percent MALES remaining aviremic’.

Response: We have now removed this sentence since this was confusingly written. These lines are indeed describing Fig 3E-H, not just Fig 3F. “Males born to recently-infected mothers were the most successful group in maintaining aviraemia (Fig 3D-H)” is the conclusion from Fig 3D and Fig 3E showing that males are more successful than females at maintaining aviraemia; from Fig 3F showing that infants (males and females) born to mothers who were recently infected are more successful than infants born to chronically-infected mothers; and from Fig 3G and Fig 3H showing the LASSO analyses also indicating these same conclusions. The y-axis of Fig 3F is correct.

We have clarified this as follows in lines 175-179:

“Although there was no sex difference in time to suppression of viraemia (<20 HIV RNA copies/ml, Fig 3C), maintenance of plasma aviraemia was significantly superior in males (Fig 3DE), even among those infants whose DNA viral loads were measured as 0 cpm PBMC (Fig 3D). Male or female infants born to recently-infected mothers were more successful at maintaining aviraemia than infants born to chronically-infected mothers (Fig 3F).”

Reviewer #2:

1. Given the justifications for greater susceptibility to IU infection in females – could the same arguments not hold true for intrapartum or breast-feeding infection? Are there any data in the literature that might support this? It seems likely that the features (viral, immune), or at least some, described for susceptibility for IU infection females in this study are unlikely to be different later on, unless immune maturation events substantially change some of these parameters. This could be discussed.

Response: This is an interesting point and one we have not been able to address to date due to lack of access to the relevant samples. However, in the Zvitambo study, the female:male ratios of *in utero*, *intra-partum* and *post-partum* infected infants were 1.49, 1.03 and 0.94, respectively. In the Malawi studies, the female:male ratios of *in utero* and *intra-partum* infected infants were 2.07 and 1.33. These suggest that the sex difference observed with respect to *in utero* mother-to-child transmission (MTCT) may not be as strong with respect to *intra-partum* and *post-partum* MTCT. We are planning future studies to explore this question further.

2. Among children with a history of ART non-adherence, 3 males maintained viral suppression while the 4 females showed viral rebound. The DNA viral loads are very low (mostly <10 copies/million PBMCs; also considering baseline levels which start higher) in the males, if compared for example to the spread of baseline levels of the 21 males and 48 females in Fig 3B

- they would be at the lower end of the distribution (some females also have similarly low levels). With DNA VLs serving as a marker for time to rebound, this begs the question of what the trajectories of DNA VLs were in the 4 females who rebounded. Similar graphs could be provided to view the RNA VL trajectories, DNA VLs and CD4 counts of the 4 female cases (in Supplementary), as presented for the male subjects. This would strengthen the findings if DNA VLs were also very low as found in the males. Alternatively, it might be that these particular female cases had much higher levels of HIV DNA, so increasing the probability of rebound (greater viral reservoir to begin with). This could add to further understanding underlying mechanisms involved in the different outcomes of these male and female cases.

Response: We absolutely agree that differences in the DNA viral loads might explain more rapid viral rebound in females versus males. In the 17 children shown in Fig 3D, all 17 had DNA viral loads measured as 0 copies/million pbmc, hence the differences in viral rebound were not the result of measurable differences in DNA viral loads.

We have clarified this in lines 176-178:

“...maintenance of plasma aviraemia was significantly superior in males (Fig 3DE), even among those infants whose DNA viral loads were measured as 0 cpm pbm (Fig 3D).”

3. *With the greater potential for functional cure suggested among HIV-1 infected males, it is interesting that the only case of a male child reported with durable post-treatment control (now for >10 years; infected either IU or IP) has not been mentioned (Violari et al, Nat Comm 2019). This case also had DNA VLs reported as 5 copies/million PBMCs at ART interruption which was the same when tested 9 years later. So DNA VLs are very much in the same range as the male cases highlighted in the current study. This would support to the arguments being made favouring males in post-treatment control.*

Response: This is a very good point and an unintended oversight on our part. We have now added the Violari reference to this case in the Discussion as follows (lines 258-262):

“With this in mind, the recent case of a male South African child³⁸ who has maintained functional cure for approaching a decade is noteworthy, contrasting with the female Mississippi child²¹ who eventually rebounded. Although these are anecdotal cases, they are consistent with the greater potential for functional cure among males proposed here.”

Small edits:

(i) *Reference 5 is incomplete.*

Response: This has now been corrected.

(ii) *Page 25, line 745 – “significant” should be “significance”.*

Response: This has now been corrected.

(iii) *Consistency – spelling of “aviraemic”, Fig3D for example has as “aviremic”.*

Response: This is the only instance of “aviremic” and this has now been corrected.

(iv) *Figure 2 E-F, G legend: “MoM” etc designations are used only for G, is currently stated under E-F.*

Response: This has now been corrected.

(v) *Supplementary Figure 3 is also labelled as Suppl Fig 2.*

Response: This has now been corrected.

Reviewer #3:

1. *Evidence of viral adaptation leading to these differences (as suggested by the Title) is weak. If anything, the TLR7 argument put forth by the authors suggests that viral adaptation is not at play.*

Response: We will respond to this point together with point 2 since there is overlap.

2. *I am confused by the framing of this manuscript with regards to immune related sexual selection. Theory would suggest that women are less susceptible to infection and have improved outcomes at the expense of auto-immune outcomes. The findings presented here do not support this hypothesis.*

Response: There are several points to make here:

(i) The TLR7/IFN-I response is stronger in females (Meier *et al*, *Nat Med* 2009, Souyris *et al*, *Science Imm* 2018).

(ii) This TLR7/IFN-I response is central to stronger immune responses to vaccines and infections (reviewed in Klein & Flanagan, *Nat Rev Imm* 2016, and Flangan *et al Ann Rev Cell Dev Biol* 2017).

(iii) In most cases this stronger immune response in females results in improved disease outcome in females. HIV and HCV infection in adults are cases in point. In HIV infection females have lower viral loads and a 5x greater likelihood of achieving elite control than males. Females have a 3x greater likelihood of achieving clearance of HCV. Both HIV and HCV have been treated successfully, especially in the case of HCV, using IFN-I therapy.

(iv) The stronger TLR7/IFN-I response in females is also associated with immunopathology – increased frequency of adverse events from vaccines and of autoimmune diseases such as SLE in which the prevalence is 9-fold higher in females and the role of the TLR7/IFN-I response and TLR7 dose is especially clearly demonstrated (Souyris *et al Science Imm* 2018; Deane *et al Immunity* 2007; Pisitkin *et al Science* 2006; Subramian *et al PNAS* 2006; Scofield *et al Arthritis Rheum* 2008 etc).

(v) In terms of transmission risk, adult studies suggest that IFN-I-resistant viruses are preferentially transmitted (Iyer *et al PNAS* 2017). In the SIV macaque model, administration of IFN α initially prevents systemic infection, but continued IFN α treatment enables infection with an increased reservoir size (Sandler *et al Nature* 2014). Thus, although as the reviewer suggests, a stronger TLR7/IFN-I response in females might at first sight be expected to decrease HIV transmission, the effect of a stronger IFN-I response is contextual and clearly will be highly influenced by viral IFN-I-sensitivity.

(vi) Females are at increased risk of mother-to-child transmission of HIV and of HCV, both by a factor of approximately 2:1 (7 studies in HIV, 2 studies in HCV, as cited in the manuscript).

(vii) The hypothesis that, in HIV infection, increased female susceptibility to mother-to-child transmission of HIV might be related to the stronger TLR7/IFN-I response that is observed in

female adults is strongly supported by the observation here of a significant difference in the IFN-I-sensitivity of the viruses transmitted *in utero* to females and males; specifically, the selection of more IFN-I-resistant viruses transmitted to females.

The data presented in the paper show strong selection of low viral replicative capacity (VRC) viruses to females *in utero* (Fig 1D, $p < 0.0001$) and of IFN-I-resistant viruses to females *in utero* (Fig 2F, $p = 0.007$), compared to males. These data would indicate that viruses that have low VRC and/or that are more IFN-I-resistant are better adapted to being transmitted to females *in utero*.

That distinct innate immune responses have evolved in males and females is not at dispute. The fact that there are two distinct immune strategies implies that there are pros and cons to each. Where the balance lies in terms of the overall consequence of the stronger IFN-I signalling in females and weaker IFN-I signalling in males is highly contextual. It is not inconsistent therefore to argue that the stronger IFN-I signalling in females results in: a) an increased risk of HIV infection, especially to IFN-I-resistant viruses; as well as b) enhanced immune control of acute infection in adults; and c) increased rate of disease progression in chronic adult infection.

To address the reviewer's comments: In the revised manuscript we have removed the word 'adaptation' from the title and have replaced it with 'selection'. We have highlighted the opening paragraph the role of TLR7 dosage and IFN-I signalling in sex differences observed in immune responses to infections and vaccines. Finally, in the Discussion we have added a paragraph to discuss the apparent paradox of increased female susceptibility to HIV despite a stronger IFN-I response as follows (lines 282-293):

"The stronger IFN-I response observed in adult females might, at first sight, be expected to decrease, rather than increase, susceptibility to HIV transmission. However, studies in the SIV macaque model illustrate the complex effects of manipulations to enhance IFN-I signalling. Administration of IFN α initially prevents systemic infection, but continued IFN α treatment enables infection with an increased reservoir size⁴³. The data presented here show that females are more susceptible to IFN-I-resistant virus, but that males are more susceptible to IFN-I-sensitive virus (Fig 2F). Thus, in a setting where mothers are recently infected and the majority of viral quasispecies in acute infection are IFN-I-resistant, as suggested by some¹⁶⁻¹⁷, but not other⁴⁴, adult transmission studies, females would be especially susceptible to *in utero* MTCT, as observed here (Fig 2AB). Further studies will be needed to evaluate whether a high frequency of IFN-I-resistant virus in acute adult infection plays a part in increased female susceptibility to *in utero* MTCT in the setting of maternal infection during pregnancy."

3. The discussion of these results with regards to "functional cure" are premature (Page 5, line 137). There is no evidence that immediate treatment of perinatally infected infants leads to a functional cure (e.g., the Mississippi baby). Also, see the Abstract, Page 2, line 61.

Response: We agree that immediate treatment does not necessarily lead to functional cure. However, there is strong evidence that a low DNA load is the best predictor of post-treatment control (Sharaf *et al* JCI 2018). There is also strong evidence that early initiation of ART results in low DNA viral loads, particularly in *in utero*-infected infants receiving ART from birth (Garcia Broncano *et al*, *Sci Transl Med*, 2019). Recent data from the SIV-macaque model, including that of perinatal infection, show that early treatment with ART can indeed achieve functional cure (Akoye *et al*, *Nat Med*, 2018; Shapiro *et al* *Nat Comm*, 2020). The high-frequency of ART interruption in infected children as a result of non-adherence provides the opportunity to assess

whether children with low DNA viral loads observed in the current cohort remained aviraemic despite this high level of ART non-adherence. The findings that, in this setting, ART non-adherent females did not maintain aviraemia for as long as males, even among infants with DNA viral loads of 0 copies/million pbmc, indicates that there is a sex difference in post-treatment control/functional cure.

To clarify the preliminary nature of these findings, we have added the following to the paragraph in the Discussion section describing the lower DNA loads and improved outcome in male infants (lines 262-264):

“Further studies involving planned analytical treatment interruption, as opposed to unplanned ART non-adherence, will be necessary to evaluate more fully the impact of sex differences on functional cure potential.”

4. I am dubious of the relevance of the competition experiments with NL4-3, a subtype B lab adapted strain from the 1980s. I know that these direct competition experiments are not rare in the field. However, competing NL4-3 against subtype C (presumably based on Fig. 4C) virus is of limited value.

Response: These are not competition experiments (in which viruses are co-cultured in competition with one another). NL4-3 is used simply as a reference against which to compare replication of other viruses. It does not matter what the clade of the reference sequence is. NL4-3 has been used as a reference sequence for comparison of viruses of non-B clade as well as B clade in numerous previous publications*. The majority of these studies used the replication capacity of NL4-3 as the reference sequence to compare non-B clade viruses. In the paper by Kiguoya *et al*, the replication capacity of NL4-3 was used as the reference sequence to compare replication capacity of A, B, C and D clade viruses.

*Miura *et al JV* 2009a, Miura *et al JV* 2009b, Miura *et al JV* 2010, Brockman *et al JV* 2010, Wright *et al JV* 2010, Wright *et al JV* 2011, Wright *et al JV* 2012a, Wright *et al JV* 2012b, Huang *et al PLoS ONE* 2011, Boutwell *et al JV* 2013, Nomura *et al JV* 2013, Juarez-Miolina *et al JV* 2014, Payne *et al PNAS* 2014, Adland *et al PLoS Path* 2015, Naidoo *et al JV* 2017, Kiguoya *et al JV* 2019, Adland *et al Bioarchiv* 2019.

5. Elite Controllers (as first referenced on Page 3, line 73) are often a result of deficient virus, rather than primarily driven by host factors (see PMIDs: 15000695 & 20504921).

Response: It is true that control of viraemia, and therefore elite control, is influenced by the replicative capacity of the transmitted virus. However, it is well-established that host factors play a very large part in elite control. In the first paper cited by the reviewer, PMID 15000695 by Yue *et al JV* 2012, two independent factors identified that contribute to early viral setpoint are gender of the recipient and HLA class I type of the recipient. In the second paper cited by the reviewer, PMID 20504921 by Miura *et al JV* 2010, using the same Gag-Pro assay as we have used here, the paper makes the point that the virus in acute infection has lower VRC in controllers compared with non-controllers. The paper goes on to show that in some cases this is because escape mutants are selected in acute infection by individuals with protective HLA alleles such as HLA-B*57. In a previous paper by the same author, using the same assay, Miura *et al (JV* 2009) showed that the HLA-B*57 subjects who become elite controllers select the particular HLA-B*57-associated escape mutants that reduce VRC the most.

The majority of elite controllers express protective HLA class I alleles such as HLA-B*57 (Pereyra *et al JID* 2008; Leslie *et al JV* 2010; Pereyra *et al Science* 2010). Other genetic factors contributing to the immune response are KIR alleles and HLA-KIR combinations. The fact that black women have a 10-fold higher likelihood of achieving elite control compared with white men infected with the same B clade of virus (frequency 0.7% versus 0.07%, Yang *et al AIDS* 2017), is further evidence that host factors play a central role in elite control.

Minor Comments:

6. *Figure 4C. This tree needs a scale bar indicating substitutions/site. The nucleotide substitution model implemented in PhyML should be provided. Also, why are there negative branch lengths? The root length obscures the tree structure. Please adjust in FigTree. Finally, both PhyML and FigTree papers need to be cited, rather than just including URLs.*

Response: The scale has been added, the tree redrawn as proposed and the two papers cited.

Page 6, line 146. I believe the authors mean “maintaining aviremia”.

Response: This has been corrected.

We thank the reviewers once again for their helpful and encouraging comments. We look forward to hearing back from you in due course.

In addition to the changes itemised above in response to the reviewers' comments we have made some minor adjustments to the text as highlighted in the file “Text with changes highlighted”. We also wish to draw the reviewers' attention to an error that appeared in the original submission in Figure 1E that has now been corrected. This figure shows GFP expression in CD4 T-cells from cord blood from HIV-uninfected males and females following *in vitro* infection. As stated in the original manuscript, this figure shows significantly higher levels of *in vitro* HIV infection in cord blood from females compared to males of HIV-uninfected twins; however, the p value is 0.03 and not 0.004 as shown in the original submission.

A Source Data file has been attached as requested, entitled “Goulder et al Source Data File” that includes source data for all the figures other than Fig 4 and all the supplementary figures.

Yours sincerely,

Professor Philip Goulder FRCPCH DPhil FMedSci
Wellcome Trust Investigator

Reviewers' Comments:

Reviewer #2:

Remarks to the Author:

This study has provided some important insights into sex-related differences in in utero-acquired infection of HIV-1. The authors have very adequately addressed all concerns and suggestions made by the reviewers; this has further enhanced the manuscript. I have nothing further to add.

Reviewer #3:

Remarks to the Author:

I am satisfied with the updated manuscript and response to reviewer comments. It is clearer now that the authors are describing host sex-specific pressures placed HIV, rather than evidence of viral adaptation to men and women.